# Association between Dietary Carbohydrate Intake and Control of Blood Pressure in Patients with Essential Hypertension

**DOI:** 10.3390/healthcare10112245

**Published:** 2022-11-09

**Authors:** Yiqing Jiang, Qin Shen, Haiying Tang, Yuanyuan Liu, Yang Ju, Ting Liu, Lingling Cui, Jingjing Li, Xiaohua Wang

**Affiliations:** 1Division of Geriatrics, The First Affiliated Hospital of Soochow University, Suzhou 215006, China; 2Outpatient Department, The First Affiliated Hospital of Soochow University, Suzhou 215006, China; 3Nursing Department, The First Affiliated Hospital of Soochow University, Suzhou 215006, China; 4Department of Orthopaedics, Shanghai JiaoTong University Affiliated Sixth People’s Hospital, Shanghai 200020, China; 5Department of Rheumatology and Immunology, Shenzhen People’s Hospital, The Frist Affiliated Hospital of Southern University of Science and Technology, Shenzhen 518000, China

**Keywords:** carbohydrate intake, blood pressure control, essential hypertension, nutrients

## Abstract

Background: Both high and low percentages of carbohydrate diets were associated with increased mortality and new-onset hypertension. However, few studies have aggregated to explore the association between carbohydrate intake and blood pressure (BP) control in patients with hypertension. This study aimed to explore the association between carbohydrate-to-energy proportion (CEP) and the rate of poorly controlled BP in patients with hypertension. Methods: A cross-sectional survey was conducted in one comprehensive hospital and one community clinic in China. Dietary CEP was obtained through two-24 h dietary recalls. According to the quintiles of CEP, the participants were divided into Q1–Q5 groups. The average of two BP values was adopted as the final BP value, and poorly controlled BP was defined as systolic BP (SBP) ≥140 mmHg and/or diastolic BP (DBP) ≥ 90 mmHg. Results: A total of 459 hypertensive patients were recruited. In univariate analyses, CEP was associated with the control of SBP and DBP. After the covariates were adjusted for, fewer CEPs in Q1 (OR, 4.335; 95% CI, 1.663, 11.299) and Q2 (OR, 2.482; 95% CI, 1.234, 4.989) were significantly associated with higher rates of poor SBP control. Conclusions: A lower dietary CEP is a risk factor for SBP control, whereas an appropriate CEP of 56% to 66% is beneficial for BP control in patients with essential hypertension.

## 1. Introduction

By 2019, the worldwide prevalence of hypertension was estimated at 34.0%. Among this population, blood pressure (BP) of only 31.7% was controlled to <140/90 mmHg [1]. The China Patient-Centred Evaluative Assessment of Cardiac Events (PEACE) study of 1.7 million patients with hypertension suggested the prevalence of hypertension in China was 37.2%, and the rate of controlled BP was only 5.7% [2]. Poor blood pressure (BP) control may cause a variety of complications such as myocardial infarction (MI), stroke, and chronic kidney disease (CKD), which also lead to increased emergency department visits, hospitalization, and healthcare costs [3]. According to statistics, the total cost of hospitalizations for hypertension in China in 2019 was as high as $2.3 billion US dollars [4]. In addition, elevated BP, especially raised systolic BP (SBP), remains the leading risk factor contributing to global mortality, accounting for 10.4 million deaths [5]. Effective BP control in patients with hypertension would lead to 803,000 fewer CVD events per year and 1.2 million quality-adjusted life years gained [4]. Therefore, BP control should be focused on the management of hypertension.

A healthy diet plays an important role in the prevention and management of hypertension [6,7,8]. Carbohydrates, as an important element of diet, are important sources of energy and account for more than half of Chinese daily caloric intake [9,10]. The health effect of carbohydrate intake is an ongoing scientific interest over the years. Evidence suggests that a low-carbohydrate diet (LCD), in which some carbohydrate is replaced by greater intake of fat and/or protein, can be an effective strategy for managing diabetes, given carbohydrate’s beneficial effects on weight loss and glycemic control [11,12]. However, several randomized trials and observational studies found that total carbohydrate intake was independently associated with BP or cardiovascular disease risk factors, although the findings were controversial [13,14,15,16,17,18]. A systematic review and meta-analysis of LCD intervention studies [13] showed that LCD had a favorable effect on BP control in overweight or obese groups. Meanwhile results from another systematic review and network meta-analysis of randomized trials suggested that the effects of LCD on the improvement of BP and other cardiovascular risk factors largely disappear at 12 months [14]. Recently, Seidelmann et al. [15] summarized that the association between carbohydrate intake and all-cause and cardiovascular mortality was nonlinear, among which both high and low carbohydrate-to-energy proportion (CEP) diets were associated with increased mortality. Similarly, a nationwide cohort study in China including 12,177 participants showed a U-shaped association between CEP and new-onset hypertension in adults, with minimal risk at dietary CEP of 50% to 55% [16]. Therefore, a certain range of CEP might be beneficial for BP control. However, to our knowledge, few studies have been conducted to investigate the association between dietary CEP and poor BP control among patients with hypertension. The purpose of this survey is to initially understand the association between CEP and the rate of poorly controlled BP, as well as to find out the reasonable range of CEP on better BP control in patients with hypertension.

## 2. Materials and Methods

### 2.1. Study Design, Inclusive and Exclusive Criteria and Ethics

This study was a cross-sectional survey. Convenience sampling was used to enroll hypertensive participants who met the inclusive and exclusive criteria from Jul. 2019 to Oct. 2020 in the Jin-chang community and the first affiliated hospital of Soochow University in China. Based on the rate of poorly controlled BP of 46.7% in Suzhou in Qi et al.’s study [19], bilateral α = 0.05, the relative error ε = 10%, using PASS 15.0 software for sample size prediction, a sample size of 458 was predicted [20,21]. The inclusions were the following: patients (1) whose age was 18 years or older; (2) who were diagnosed with essential hypertension according to the standard of the latest guidelines for the prevention and treatment of hypertension in China [8]; (3) who provided informed consent. Patients who had the following conditions were excluded: (1) secondary hypertension; (2) severe physical comorbidities or complications (e.g., cancer, severe heart, kidney, and liver failure); (3) lactating or pregnant women; (4) cognitive dysfunction; (5) participation in other research. This study was conducted in accordance with the declaration of Helsinki and with approval from the Ethics Committee of the First Affiliated Hospital of Soochow University (No. ECSU-2019000148). Written informed consent was obtained from all participants.

### 2.2. The Blood Pressure

After finishing the survey of demographic and clinical data, the patient rested for at least five minutes. Using the corrected OMRON sphygmomanometer (HEM-8102), the office BP (oBP) was evaluated in the patient’s upper arm at a sitting position in a community service room or a teaching room of the hospital. A repeated oBP measurement was performed after a five-minute interval. The average of two oBP values was adopted as the final BP value. Poorly controlled BP was defined as an average SBP ≥ 140 and/or DBP ≥ 90 mmHg [7].

### 2.3. Diet Record and Definition of Carbohydrate-to-Energy Proportion

The data on carbohydrate and other nutrient intake were assessed using two 24 h diet records (a weekend and a day on a weekday) [22]. Six trained researchers instructed the participants to record detailed dietary intake over 2 days (one day before the survey face-to-face and the second or third day after the survey by Wechat or telephone). The Feihua Nutrition Software (V2.7.6.10, Beijing, China) was applied to calculate the quantities of nutrients ingested. The average amount of nutrients in two recording days was adopted. In this study, the CEP was defined as the energy provided by carbohydrates divided by the total energy, then multiplied by 100. Based on the study by Li et al. [16], the subjects were divided into five groups by the quintiles of CEP from low to high: Q1, Q2, Q3, Q4, and Q5, respectively.

### 2.4. Demographic and Clinical Data

A general information questionnaire covering demographic and clinical data was developed by the team. The demographic data included age, gender, educational degree (below high school/high school or above), medical insurance (yes/no), regular exercise (yes/no), duration of sleep (h/d), quality of sleep (fair/poor), smoking and alcohol (yes/no), body mass index (BMI), duration of hypertension (y), taking antihypertensive drugs (yes/no), complication (yes/no), comorbidity (yes/no), and family history of hypertension (yes/no). Regular exercise was defined as exercising ≥ 3 times a week, ≥ 20 min each time, and continuous time ≥ 3 months [23]. The evaluation of the quality of sleep was self-reported using a VAS scale of 10 scores (≤ 3 poor, >3 not poor) [24]. Patients’ weight and height were measured with participants wearing light clothes and no shoes, and BMI was calculated as weight (kg) divided by the square of height (m). According to the Guidelines for the Prevention and Control of Overweight and Obesity in Adults in China [25], a BMI≥ 28.0 was defined as obesity. Hypertensive complications included left ventricular hypertrophy, heart failure, myocardial infarction, frequent angina, aortic dissection, stroke, cerebral hemorrhage, and renal insufficiency [26]. Comorbidity means the co-existence of one or more diseases or clinical conditions and is independent of BP [27]. When necessary, the researcher helped participants determine whether a condition is a complication or a comorbidity.

### 2.5. Data Collection Process

The data collection process was as follows. (1) The researcher informed participants of the purpose of the study, and participants who met the inclusion criteria signed an informed consent form. (2) The participant filled in the general questionnaire, which contained demographic and clinical data and took about ten minutes. (3) Then, the participant’s parameters were measured, including oBP. (4) The participant’s detailed diet data one day before the survey were recalled and recorded. (5) The researcher obtained the participant’s contact method (WeChat or cellphone numbers). (6) On the second or third day after the survey, the researchers obtained the second 24 h dietary data through WeChat or telephone.

### 2.6. Statistical Analysis

Statistical analyses were performed using SPSS 25.0 software (SPSS, Inc., Chicago, IL, USA). The analysis mainly included the following aspects:(1)Description of demographic, clinical, and nutritional data: for categorical variables, the results were described as the frequency (percentages); for continuous variables, we determined if data were normally distributed using the Kolmogorove–Smirnov test before analysis. If normal, they were expressed as mean ± SD; otherwise, data were expressed as median (P_25_, P_75_). Comparisons of demographic, clinical, and nutritional data between the two groups: for categorical variables, Pearson’s Chi-square test or Fisher’s exact test was used; for continuous variables, if they were normally distributed, the comparison between groups was made using the independent samples *t* test; otherwise, the Manne–Whitney U test was used.(2)Pearson’s Chi-square test was used to initially analyze the relationship between the CEP and the rates of poorly controlled BP.(3)In order to adjust for the impact of covariates on BP, multivariate binary logistic regression was used to analyze the associations between the CEPs and the rates of poorly controlled BP. The covariates were those indicators with *p* < 0.10 in the demographic, clinical, and nutritional data.(4)A *p* value of <0.05 was considered statistically significant.

## 3. Results

### 3.1. Demographic and Clinical Characteristics

Initial questionnaires were collected from a total of 471 participants. Among those collected, the 459 were considered valid, and the effective recovery rate was 97.5%. The mean age of participants was (51.31 ± 12.62) years, and 288 (62.75%) were men. The mean duration of hypertension was four (0.5,10.0) years. Participants’ socio-demographic and clinical characteristics are presented in Table 1. Compared with those with good SBP control, participants with poor SBP control were younger, with a shorter duration of hypertension and higher BMI, less likely to take antihypertensive drugs, and had more cholesterol intake; whereas DBP-related variables included being younger, having higher education, being male, alcohol drinking and smoking, higher BMI, with a shorter duration of hypertension, and less likely to take antihypertensive drugs.

### 3.2. The Status of Dietary Carbohydrate, Fat, and Protein Intake

Among all participants, the mean dietary carbohydrate intake was (286.12 ± 79.22) g/day, and the median dietary CEP was 58.53%. The subjects were divided into five groups by the quintiles of CEP. The median percentages of CEP in Q1, Q2, Q3, Q4, and Q5 were 40.99%, 53.14%, 58.53%, 62.99%, and 69.48%, respectively; the fat-to-energy proportions were 43.1%, 32.0%, 27.9%, 24.6%, and 26.6%, respectively; and the protein-to-energy proportions were 16.2%, 16.2%, 13.3%, 13.3%, and 14.3%, respectively.

### 3.3. Statuses of BP Control

Among the patients with hypertension in our study, the mean of SBP and DBP were (136.37 ± 13.49) mmHg and (86.28 ± 11.21) mmHg, respectively. The number (rate) of patients with poor BP control was 237 (51.63%). There were 171 (37.25%) poorly controlled SBP cases, with a median BP of 146.00 mmHg, and 179 (39.00%) poorly controlled controlled cases, with a median BP of 96.00 mmHg.

### 3.4. Comparison of the Rates of Poorly Controlled BP in Groups with Different CEPs

The rates of the poorly controlled SBP and DBP in each group with different CEPs are presented in Table 2. There was a J-shaped relationship between the dietary CEPs and the rates of poorly controlled SBP and a skewed U-shaped relationship with DBP. The poorly controlled rates of SBP and DBP from Q1 to Q4 gradually decreased. Compared with Q4, the Q5 group showed an increase in the rate of poorly controlled DBP. The differences in the rates of poorly controlled SBP and DBP among the five groups were statistically significant (SBP: χ^2^ = 20.767, *p* < 0.001; DBP: χ^2^ = 10.441, *p* = 0.034). Q4 had the lowest poorly controlled rates of SBP and DBP (25.81% and 30.11%, respectively). See Table 2.

### 3.5. Associations between Dietary CEPs and Control of BP

In order to explore the associations between CEP and BP, a binary logistic regression was performed, with the CEP as an independent variable, the demographics, clinical indicators, and nutrient intake (*p* < 0.10) as covariates, the controlled rates of SBP and DBP as the dependent variables, respectively; taking Q4 (the lowest rates of poor controlled BP) as a reference group, the results showed that the fewer dietary CEPs in Q1 (OR, 4.335; 95% CI,1.663, 11.299) and Q2 (OR, 2.482; 95% CI, 1.234, 4.989) were significantly associated with the higher poorly controlled rates of SBP, whereas there were no statistically significant associations with those in other CEPs groups; and there were no statistically significant associations between dietary CEPs and the poorly controlled rates of DBP. See Table 3.

## 4. Discussion

This cross-sectional survey investigated the association between the percentage of energy from carbohydrate intake and poor BP control in patients with essential hypertension. The results indicated that lower CEP was associated with a higher risk of poor SBP control, with the highest rate of poorly controlled SBP (50.55%) in the group with <49% of CEP, whereas the lowest risk of poorly controlled SBP (25.81%) was in the group with 61% to 66% of CEP. In addition, those with poor BP control in this study were younger, had a shorter duration of hypertension and higher BMI level, consumed more cholesterol, and did take antihypertensive medications, which is consistent with several previous studies [7,28].

### 4.1. Status of Carbohydrate Intake

In the traditional Chinese diet, staple foods are mainly coarse grains; therefore, carbohydrate intake accounts for a high proportion of energy (mean values about 60–70%) [29,30]. In this study, the mean daily intake of carbohydrates was (286.12 ± 79.22) g/d, and the mean CEP was (57.15 ± 11.26) %, which is consistent with the findings of the China National Nutrition Surveys (CNNS) [31]. This means that CEP is basically accordant with the recommended amount specified in the currently acceptable macronutrient distribution range (AMDR) in China (for adults, 50% to 65%) [32]. In addition, we found that approximately 40% of patients had a dietary CEP of less than 56%, and 20% had a CEP above 66%, which means that their carbohydrate intake might not be within a reasonable range.

### 4.2. Status of Poorly Control of BP

Unhealthy dietary habits contributed to poorly controlled BP [33,34]. The present study found that the rates of patients with poor BP, SBP, and DBP control were 51.63%, 37.25%, and 39.00%, respectively, which is similar to the results by Ke et al. [28], who found the rate of poor BP control was 51% in Macau, China. However, it is lower than the findings of Qu’s study [35] conducted in community medical centers of Liaoning province, a northeast district of China, which showed that the rate of poor BP control was 67.1%. This difference may be explained by the different study settings. Studies showed there is more poor BP control in the less economically developed districts of southwestern and northeastern provinces of China [36]. In addition, patients in hospitals generally have a high level of health awareness, which contributes to good compliance with health behaviors and is thus beneficial to the control of BP. Most of the participants in this study were recruited from hospitals in Suzhou, a more economically developed city.

### 4.3. Association between Dietary CEP and Poor SBP Control

Elevated SBP, as a significant predictor of cardiovascular events after age 50 [37,38], accounts for the majority of the burden of death and disability due to hypertension [39]. Carbohydrate-rich foods (e.g., grains, legumes, vegetables, and sugary beverages) may affect BP control through mechanisms related to obesity, the insulinemic response pathway, and vascular dysfunction [40,41]. Long-term consumption of diets low in carbohydrates is associated with an increase in total mortality [15]. However, the association between carbohydrate intake and the onset and development of hypertension remains unclear. Song et al. surveyed 14,438 adults in Korea and found that men in the highest quintile of carbohydrate intake (g/day) compared to the lowest quintile of the group had a 34% lower risk of elevated BP [17]. Similarly, a cross-sectional study of healthy Chinese adolescents surveyed by Zhu et al. also showed that male adolescents consuming a high-CEP (≥55%) diet had lower SBP than those consuming a non-high CEP (<55%) diet [42]. This study showed that CEP was an independent risk factor for SBP control, that is, a lower CEP was associated with a higher risk of poor SBP control, which is similar to the findings of the above studies. Compared with that of the groups with a CEP of 61 % to <66%, the poorly controlled rates of SBP in those groups with a CEP of <49% and 49% to <56% significantly increased by 4.335 and 2.482 times, respectively. The reasons for the results may be that the participants who consumed a low CEP were more likely to consume more fat (43.09% and 32.03% in the Q1 and Q2 groups), especially saturated fat (SFA), which may elevate BP through a proinflammatory mechanism within the endothelium [43]. In recent years, with economic and cultural development, China has experienced a shift from traditional to Western dietary patterns, with a decrease in cereal and vegetable consumption and an increase in meat and packaged food consumption [10,44]. This finding also implies that the quality of protein and fat replacing carbohydrates in LCD is an important consideration. In addition, some researchers have noted that people who follow LCD generally have a lower consumption of foods such as fruits, vegetables, and whole grains, which may put them at risk for nutrient deficiencies such as dietary fiber [45]. Dietary fiber can increase the production of short-chain fatty acids (SCFAs) in the intestine by regulating intestinal flora [46,47]. SCFAs can inhibit renin secretion and decrease RAAS activity, thereby improving BP [48,49]. Therefore, the consumption of a lower CEP diet, which may be detrimental to BP control, should raise concerns for patients with hypertension.

### 4.4. Association between Dietary CEP and Poor DBP Control

A prospective study of 1.3 million people showed that diastolic hypertension is also an independent predictor of adverse cardiovascular outcomes, although it is often overlooked [50]. Less carbohydrate intake may also be risk factor for poor DBP control; the reason may be that a low CEP diet is associated with a significant increase in low-density lipoprotein cholesterol (LDL-C) [51], which is associated with increased systemic vascular resistance and leads to increased DBP [52,53]. On the contrary, high carbohydrate intake may cause hyperglycemia, which, through the mediator of oxidative stress, may allow abnormal endothelium-dependent arterial diastole and therefore increase DBP [18,41]. In this study, the relationship between CEP and the rate of poorly controlled DBP in the univariate analysis was demonstrated with a skewed U-shaped curve. There was only about a 30% poor DBP control rate in the group with a range of 56–66% CEP, whereas there were 38% and about 50% poor DBP control rates in the groups of CEP ≥ 66% and <56%, which is consistent with the results of Soh’s study [18], in which a nationally representative cross-sectional survey in Korea indicated that a low (<45%) or high (≥70%) energy intake from carbohydrates were associated with incidences of hypertension. However, after adjusting for covariates, there was no statistically significant association between high CEP and poor control of DBP, which may be because participants in the high CEP group followed the Chinese southern traditional diet characterized by healthful carbohydrate-rich food items such as whole grains, fruits, vegetables, and legumes, which may contribute to improving DBP control [54].

Currently, it is well-known the Mediterranean and Dietary Approach to Stop Hypertension (DASH) diets are commonly recommended for patients with hypertension. These dietary regimens are not overly restrictive on carbohydrates, but rather make recommendations for healthier, more carbohydrate-rich foods [55,56]. Combined with our findings, LCD may not be an effective strategy for managing hypertension. In the future, the impact of carbohydrate quality on BP control should be a focus.

### 4.5. Limitations

There are some limitations to this study. First, we did not investigate the source and quality of carbohydrates, since dietary carbohydrates consist of sugars, starches, and fiber, which can affect health outcomes in different ways. Savoia et al. [57] also highlighted the important role of carbohydrate quality in the development of hypertension and cardiovascular disease, which should be a focus in our future studies. Second, this study failed to analyze the effects of SFAs, unsaturated fatty acids, and protein–carbohydrate interactions on poor BP control. Third, we did not take into account the intake of supplements, and we will include this variable as a confounding factor in future studies, Fourth, we did not take psychological factors as confounding variables to exclude the influence of psychological participation in BP control. Fifth, this study was conducted in Suzhou, a more economically developed city in China, but we did not include the location as a confounding factor. Thus, the generalizability of the findings has some limitations. Finally, it is a cross-sectional survey that cannot explain causality; more prospective cohort studies and randomized controlled trials are still needed to confirm our findings. Therefore, the optimal intake range for energy proportion of carbohydrate intake in the hypertensive population still needs to be further explored.

## 5. Conclusions

A lower dietary CEP is a risk factor for SBP control, whereas an appropriate CEP of 56% to 66% is beneficial for BP control in patients with essential hypertension. In this study, approximately 40% of patients with essential hypertension had a dietary CEP of less than 56%. Considering that carbohydrates are an important part of the diet of Chinese residents, these insights should be taken into account when formulating rational dietary programs for hypertensive patients.

## Figures and Tables

**Table 1 healthcare-10-02245-t001:** Socio-demographic and clinical characteristics (*n* = 459).

Characteristics	SBP $\overset{-}{x}$ ± s/M (P_25_,P_75_)/n (%)				DBP $\overset{-}{x}$ ± s/M (P_25_,P_75_)/n (%)			
	Good Control (*n* = 288)	Poor Control (*n* = 171)	t/χ^2^/z	*p* Value	Good Control (*n* = 280)	Poor Control (*n* = 179)	t/χ^2^/z	*p* Value
Age (y)	53.23 ± 12.21	48.06 ± 12.68	4.319 ^a^	<0.001 ***	54.60 ± 12.18	46.15 ± 11.57	7.393 ^a^	<0.001 ***
Sex (male)	180 (62.50)	108 (66.67)	0.020 ^b^	0.888	164 (58.57)	124 (69.27)	5.350 ^b^	0.021 *
Education (≥high school)	140 (48.61)	69 (40.35)	2.952 ^b^	0.086	151 (53.93)	58 (32.40)	20.402 ^b^	<0.001 ***
Medical insurance (no)	12 (4.17)	12 (7.02)	2.200 ^b^	0.333	13 (4.64)	11 (6.15)	0.497 ^b^	0.481
Regular exercise (no)	213 (73.96)	126 (73.68)	0.004 ^b^	0.948	208 (74.29)	131 (73.18)	0.069 ^b^	0.793
Duration of sleep (h)	6.86 ± 1.17	6.82 ± 1.06	0.323 ^a^	0.747	6.86 ± 1.17	6.82 ± 1.06	−0.943 ^a^	0.346
Quality of sleep (poor)	21 (7.29)	16 (9.36)	0.617 ^b^	0.432	23 (8.21)	14 (7.82)	0.023 ^b^	0.880
Smoking (yes)	71 (24.65)	47 (27.49)	0.451 ^b^	0.502	58 (20.71)	60 (33.52)	9.375 ^b^	0.002 **
Alcohol drinking (yes)	65 (22.57)	45 (26.32)	0.826 ^b^	0.363	51 (18.21)	59 (32.96)	13.031 ^b^	<0.001 ***
BMI	24.9 (22.6, 27.0)	25.2 (23.6, 27.5)	−2.066 ^d^	0.039 *	24.6 (22.5, 26.8)	25.6 (23.8, 27.8)	−3.572 ^d^	<0.001 ***
Obesity (yes)	43 (14.9)	34 (19.9)	1.885 ^b^	0.170	35 (12.5)	42 (23.5)	9.401 ^b^	0.002 **
Duration of HTN (y)	5.5 (1.0,11.0)	2.0 (0.4, 6.5)	−4.479 ^d^	<0.001 ***	5.0 (1.0, 10.0)	2.0 (0.5, 6.8)	−3.593 ^d^	<0.001 ***
Antihypertensive drugs (no)	49 (17.01)	55 (32.16)	14.053 ^b^	<0.001 ***	43 (15.36)	61 (34.08)	21.839 ^b^	<0.001 ***
Complication (yes)	3 (1.04)	2 (1.17)	0.000 ^c^	1.000	4 (1.43)	1 (0.56)	0.172 ^c^	0.678
Comorbidity (yes)	39 (13.54)	23 (13.45)	0.001 ^b^	0.978	41 (14.64)	21 (11.73)	0.792 ^b^	0.373
Family history of HTN (yes)	241 (83.68)	139 (81.29)	0.432 ^b^	0.511	230 (82.14)	150 (83.80)	0.210 ^b^	0.647
PEP (%)	14.75 ± 3.19	14.89 ± 3.67	−0.380 ^a^	0.704	14.51 ± 3.13	15.25 ± 3.68	−2.311 ^a^	0.021
FEP (%)	30.78 ± 8.09	32.12 ± 8.10	−1.714 ^a^	0.087	30.80 ± 7.97	32.03 ± 8.30	−1.591 ^a^	0.112
Cholesterol (mg/d)	297 (175, 427)	350 (198, 482)	−2.227 ^d^	0.026 *	297 (178, 439)	350 (189, 457)	−1.643 ^d^	0.100
Sodium (mg/d)	2109 (1395, 2583)	2108 (1770, 2500)	−1.351 ^d^	0.177	2173 (1602, 2593)	2013 (1631, 2465)	−1.343 ^d^	0.179
Potassium (mg/d)	1674 (1474, 1935)	1644 (1311, 1894)	−0.971 ^d^	0.332	1662 (1365, 1898)	1704 (1378, 1936)	−0.988 ^d^	0.323

Notes: ***: *p* < 0.001; **: *p* < 0.01; *: *p* < 0.05. HTN: hypertension; BMI, body mass index; SBP, systolic blood pressure; DBP, diastolic blood pressure; PEP, protein-to-energy proportion; FER, fat-to-energy proportion. ^a^ Independent-samples *t* test; ^b^ Pearson’s Chi-square; ^c^ Fisher’s exact test; ^d^ Mann–Whitney U. M (P25, P75), median (25th and 75th percentiles).

**Table 2 healthcare-10-02245-t002:** The rates of poor controlled blood pressure according to CEPs.

Group (CEP, %)	SBP			DBP		
	Poor Control N (%)	χ^2^	*p*	Poor Control N (%)	χ^2^	*p*
Q1 (<49)	46 (50.55)	20.767	<0.001 ***	44 (48.35)	10.441	0.034 *
Q2 (49 to <56)	44 (47.31)			43 (53.76)		
Q3 (56 to <61)	33 (35.26)			29 (31.87)		
Q4 (61 to <66)	24 (25.81)			28 (30.11)		
Q5 (≥66)	24 (26.37)			35 (38.46)		

Notes: CEP, carbohydrate-to-energy proportion; Q, wuintile; Q1, P_0_−P_20_; Q2, P_20_−P_40_; Q3, P_40_−P_60_; Q4, P_60_−P_80_; Q5, P_80_–P_100_. ***: *p* < 0.001; *, *p* < 0.05. SBP, systolic blood pressure; DBP, diastolic blood pressure.

**Table 3 healthcare-10-02245-t003:** The associations between the dietary CEPs and poor controlled rates of SBP and DBP (adjusted for covariates).

Variables	SBP			DBP		
	B	*p*	OR (95%CI)	B	*p*	OR (95%CI)
Q4	-	-	1.000 (Ref.)	-	-	1.000 (Ref.)
Q1	1.467	0.003 **	4.335 (1.663, 11.299)	0.148	0.699	1.159 (0.548, 2.452)
Q2	0.909	0.011 *	2.482 (1.234, 4.989)	0.292	0.408	1.339 (0.670, 2.673)
Q3	0.551	0.100	1.735 (0.899, 3.348)	−0.091	0.798	0.913 (0.456, 1.828)
Q5	−0.031	0.927	0.969 (0.493, 1.904)	0.348	0.306	1.416 (0.727, 2.755)

Notes: **: *p* < 0.01; *: *p* < 0.05. SBP, systolic blood pressure. SBP, adjusted for age, sex, education degree, BMI, duration of hypertension, taking antihypertensive drugs, cholesterol intake, fat-to-energy proportion; DBP, diastolic blood pressure. DBP, adjusted for age, sex, education degree, smoking status, alcohol drinking, BMI, duration of hypertension, taking antihypertensive drugs, protein-to-energy proportion.

## Data Availability

The datasets generated during this study are available by mail to corresponding authors upon reasonable request.

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
