# Peer review of "Association between Dietary Carbohydrate Intake and Control of Blood Pressure in Patients with Essential Hypertension"

_healthcare, 2022, doi:10.3390/healthcare10112245_

Round 1

Reviewer 1 Report

The authors investigated the relationship between carbohydrate-to-energy proportion (CEP) and blood pressure control. They found that CEP<56% was associated with a higher risk of poor blood pressure control in patients with essential hypertension. It is an attractive result for improving the control rate of hypertension. I have some minor concerns:

Although the study included 471 participants, there are no obese patient including.

The authors should clearly state the reason for dividing into five groups, not more or less groups.

N should be 288, not 280 in Table 1 under DBP column.

Lines 278-282 should be re-organized. It is mis-leading statement.

Reviewer 2 Report

Major comment

The data on carbohydrate and other nutrient intake was assessed using two 24h diet records. There is no adequate assessment of dietary habits, valid determination is 24-hour diet records on two weekdays and one weekend day.

The results cannot be discussed if the intake of protein and fat in the energy share is not shown and discussed.

Reviewer 3 Report

Manuscript entitled " Association between Dietary Carbohydrate Intake and Control 2 of Blood Pressure in Patients with Essential Hypertension" is within the scope of Healthcare.

This work has interesting subject and manuscript is fairly well written but the article in its present form may not be possible for publication in Healthcare journal unless some important points bein addressed as follow:

1-     The whole English language of the article needs to be edited and my advice is to seek help from the Publisher's editing services and to ask one of the peers who is native English speaker to read the whole review article for language and punctuation adjustment.

2-     Introduction section - the Authors should try to make an effort to emphasize the importance of their studies.

3-     Discussion is very important part of each manuscript published. In presented manuscript this section comprises too general explanations. Authors should discuss their results with more others and newest scientific papers.

4-     The conclusions should reflect the innovation of this study and the perspectives.

5-     References must be corrected

 Manuscript entitled " Association between Dietary Carbohydrate Intake and Control 2 of Blood Pressure in Patients with Essential Hypertension" is within the scope of Healthcare.

This work has interesting subject and manuscript is fairly well written but the article in its present form may not be possible for publication in Healthcare journal unless some important points bein addressed as follow:

1-     The whole English language of the article needs to be edited and my advice is to seek help from the Publisher's editing services and to ask one of the peers who is native English speaker to read the whole review article for language and punctuation adjustment.

2-     Introduction section - the Authors should try to make an effort to emphasize the importance of their studies.

3-     Discussion is very important part of each manuscript published. In presented manuscript this section comprises too general explanations. Authors should discuss their results with more others and newest scientific papers.

4-     The conclusions should reflect the innovation of this study and the perspectives.

5-     References must be corrected

Reviewer 4 Report

The manuscript submitted to Healthcare to be considered for publication by Jiang et al., titled: "Association between Dietary Carbohydrate Intake and Control of Blood Pressure in Patients with Essential Hypertension" is an observational human work with potential clinical implications.

The reviewer would like to present the following points for the authors' consideration towards improvement of the manuscript:

1. The introduction would benefit from some enrichment and elaboration. It would be helpful to present more statistics both in terms of local and global levels in terms of the burdens of the problem (financial) and discuss about potential benefits from alleviating/addressing the problem.

2. Be more concise in the introduction as per the research question and aim of the work, possibly also including a clear hypothesis statement.

3. The authors state that the employed a convenience sample. While this is understandable could the location/site constitute thus a confounding factor? This needs to be addressed and discussed in the context of representability of the obtained results.

4. Along those lines what are potential confounding factors and how were they addressed/normalized/corrected for? For example the age range is considerable since the only stipulation the researchers stated was for participants to be adults. Age in terms of the pathology can be a significant variant and a notable confounding factors.

5. Was smoking status and medication or supplement administration considered as potential confounders by the authors?

6. In the results section some tables (eg table 1) are fairly congested. It would be much better for the reader and the quality of the results presentation for the tables to be redesigned so that it is easier to follow the information presented.

7. BMI is an index and does not have units. The kg/m2 pertains to the way BMI is calculated and does not represent units (the units would be meaningless since we are not measuring surface area as m2 suggests and definitely not measuring pressure or force which is what kg/m2 suggests). While unfortunately it is not uncommon to see BMI with units in the literature from a purely scientific perspective it is wrong to assign units to BMI.

8. Was the type of carbohydrate considered from a dietary perspective?There is evidence to suggest that differentiation in the type of carbohydrate can produce different microbiome outcomes and subsequently modulate CVD risk for example which includes atherosclerosis and downstream blood pressure. 

9. Was potential pre-diabetes status considered? This is also a potential effector and possibly a confounder in terms of Blood Pressure (BP) outcomes.

10 The discussion would benefit from further elaboration and broader considerations of the literature. Especially discussing potential relationships with diabetes, dietary intake etc. Here are some papers that are suggested for inclusion in such discussion:

1. Sikalidis, A.K.; Maykish, A. The Gut Microbiome and Type 2 Diabetes Mellitus: Discussing A Complex Relationship. Biomedicines 2020, 8, 8. https://doi.org/10.3390/biomedicines8010008

2. Sikalidis, A.K.; Kelleher, A.H.; Kristo, A.S. Mediterranean Diet. Encyclopedia 2021, 1, 371-387. https://doi.org/10.3390/encyclopedia1020031 

11. The manuscript would benefit from proofreading by a native English speaker so that syntax, grammar, style and flow are improved.

Reviewer 5 Report

First of all, I would like to thank you for sending me this article for review and congratulate the authors for their initiative in this highly relevant research in the field of health in schoolchildren.

In the abstract it is recommended to eliminate the numbers in parentheses to indicate each section.

The introductory sentence in the abstract does not provide any information. It should be improved.

The sample size should be explained in more detail. Based on what it was calculated.

It is recommended to use the Strobe Statement.

It is not understood why this definition of physical activity practice has been used instead of the international definition of the World Health Organization. It is recommended to perform the analyses taking into account this definition.

It is also recommended to include analyses for mean arterial blood pressure, as well as controlling for BMI.

Round 2

Reviewer 2 Report

The authors have corrected the paper in accordance with my suggestions

Reviewer 3 Report

The manuscript has been significantly improved. No further comments.

Reviewer 4 Report

The authors have made a reasonable effort in addressing reviewers' comments. 

Reviewer 5 Report

Thank you to the authors